# Synthesis and Electrochemical Energy Storage Applications of Micro/Nanostructured Spherical Materials

**DOI:** 10.3390/nano9091207

**Published:** 2019-08-27

**Authors:** Qinghua Gong, Tingting Gao, Tingting Hu, Guowei Zhou

**Affiliations:** Key Laboratory of Fine Chemicals in Universities of Shandong, School of Chemistry and Pharmaceutical Engineering, Qilu University of Technology (Shandong Academy of Sciences), Jinan 250353, China

**Keywords:** micro/nanostructures, complex hollow spheres, electrochemical energy storage

## Abstract

Micro/nanostructured spherical materials have been widely explored for electrochemical energy storage due to their exceptional properties, which have also been summarized based on electrode type and material composition. The increased complexity of spherical structures has increased the feasibility of modulating their properties, thereby improving their performance compared with simple spherical structures. This paper comprehensively reviews the synthesis and electrochemical energy storage applications of micro/nanostructured spherical materials. After a brief classification, the concepts and syntheses of micro/nanostructured spherical materials are described in detail, which include hollow, core-shelled, yolk-shelled, double-shelled, and multi-shelled spheres. We then introduce strategies classified into hard-, soft-, and self-templating methods for synthesis of these spherical structures, and also include the concepts of synthetic methodologies. Thereafter, we discuss their applications as electrode materials for lithium-ion batteries and supercapacitors, and sulfur hosts for lithium–sulfur batteries. The superiority of multi-shelled hollow micro/nanospheres for electrochemical energy storage applications is particularly summarized. Subsequently, we conclude this review by presenting the challenges, development, highlights, and future directions of the micro/nanostructured spherical materials for electrochemical energy storage.

## 1. Introduction

As a unique family of functional materials, spherical structures offer structural stability, large surface area, low density, and short charges transport lengths [1,2,3,4]. Spherical structures can be classified based on their structural complexity into simple and intricate ones. On the one hand, simple spherical structures, also known as solid spheres [5,6] and single-shelled hollow spheres. On the other hand, intricate hollow structures have multiple layers and interior cores [7,8,9]; these structures include core-shelled [10,11,12,13], yolk-shelled [14,15,16,17], double-shelled, and multi-shelled spheres. Spherical structures exhibit tunable physical and chemical properties, which confer them with great structure advantages for electrochemical applications, such as lithium-ion batteries (LIBs) [18,19,20], lithium–sulfur batteries (LSBs) [21,22,23,24], supercapacitors (SCs) [25,26,27,28], sodium-ion batteries, Li–selenium batteries, and fuel cells [29,30,31,32,33,34]. Hollow spherical micro/nanostructures with high complexity have attracted much interest for theoretical studies [35,36,37,38,39] and practical applications [40,41]. Micro/nanostructured spherical materials are expected to outperform other structures in terms of enhanced electrochemical performance and structural stability [42,43,44].

LIBs are outstanding among electrochemical energy storage technologies in terms of high energy density [45,46,47,48]. LIBs were first commercialized in 1991 by Sony Corporation [49,50,51,52]. Currently, LIBs provide a voltage of the order of 4 V, and energy density ranging from 100 to 150 Wh kg^−1^. LIBs have three main application domains: portable electronics, electric vehicle, and stationary energy storage [53,54,55]. So far, the first usage is the best developed and the largest in terms of the number of units generated. The performance of LIBs significantly depends on the active anodes, which are used to store and release Li-ions during charging and discharging. The most popular anode worldwide is graphite anode due to its stable potential, low cost, and long cycle life. However, graphite anode has a limited theoretical specific capacity of 372 mA h g^−1^ and poor rate capacity, which are insufficient for the development of portable electronic devices and EVs [56,57]. LSBs hold tremendous potential as energy storage devices due to their high theoretical specific capacity (1675 mA h g^−1^), and energy density (2600 Wh kg^−1^) [58]. Since 2009, LSBs have received increasing attention and are considered as one of the most promising candidates for next-generation rechargeable batteries. From the recent improvements in the Li–S system, it seems that the practical application of LSBs is not far away. However, Li–S cells have hindrances in their commercial application due to their limited conductivity, volume expansion, and rapid capacity fading [59]. The earliest SCs patent was filed in 1957. However, not until the 1990s did SCs technology begin to draw some attention, in the field of hybrid electric vehicles [60,61,62]. It was found that the main function of SCs could be to boost the battery in a hybrid electric vehicle providing the necessary power for acceleration, with an additional function being to recuperate brake energy [63,64,65,66]. Further developments have led to the recognition that SCs can play important roles in complementing batteries or fuel cells in their energy storage functions by providing back-up power supplies to protect against power disruptions [67]. As a result, the US Department of Energy has designated SCs to be as important as batteries for future energy storage systems [68]. Recent years, major progress have been yielded in the theoretical and practical research and development of SCs, as evinced by a large number of research articles and technical reports. With the development in backup power sources, portable electronics devices, renewable energy power plants, and EVs, further improvement in energy and power density for SCs is imperative. The key objective is to fabricate outstanding electrode materials with large specific capacitance, high power delivery, and good cycling stability [69,70]. 

In the specific field of electrochemical energy storage, spherical structures are playing a more and more important role. More importantly, they hold great promise to break some of the current bottlenecks in LIBs, LSBs, and SCs [8,32]. For example, the spherical structures offer structural stability, and the cavity of intricate hollow spherical structures can effectively accommodate the volume change of high-capacity LIBs anode materials and boost the cycling stability. In LSBs, such unique structures can reserve a large amount of S, accommodate the volume variation of S during cycling, and avoid the discharged products from dissolution through either physical confinement or chemical interactions. Also, in SCs, spherical structures can generally increase the energy densities of energy-storage devices due to their large surface area, low density, and high weight fraction of active species.

In this review, we mainly summarized the latest development on the micro/nanostructured spherical materials including the typical structural types, and their applications for energy-related. The examples we enumerated in this review are the typical representatives in terms of the micro/nano-architectures related to the energy applications. Scheme 1 shows that these micro/nanostructured spherical materials are categorized into hollow, core-shelled, yolk-shelled, double-shelled, and multi-shelled micro/nanospheres. Their applications as the sulfur hosts for LSBs, electrode materials for LIBs, and SCs conversion reactions are then discussed. Subsequently, the challenges, development, highlights, and future directions of micro/nanostructured spherical materials are concisely given.

## 2. Synthesis of Micro/nanostructured Spherical Materials 

### 2.1. Hollow Spheres with Complex Architectures

Hollow spheres are simple spherical structures with narrow size distribution and superior morphological uniformity, and usually obtained through template methods, such as hard- and soft-templating. The mechanism of formation process, transmission electron microscopy (TEM), and scanning electron microscopy (SEM) images of hollow spheres are illustrated in Figure 1.

Hard-templating is the common method, and the hard template can be selectively removed through etching or annealing. Typically, the desired materials or their precursors are deposited on hard templates with functionalized surface, followed by the selective removal of the templates through etching or pyrolysis. A myriad of inorganic/organic colloidal spheres could be applied as ideal hard templates (SiO_2_, polystyrene (PS), and so on) owing to their facile preparation [71,72]. SiO_2_ spheres with controllable size can be easily prepared by the Stöber process on a large scale. Therefore, we select two typical examples of using SiO_2_ spheres as the hard templates. For instance, Lou et al. synthesized mesoporous TiO_2_ hollow spheres through SiO_2_ spheres as hard template, hexadecylamine (HDA) structure-directing agent, and titanium isopropoxide (TIP) as the TiO_2_ precursor, respectively [73]. The synthesis strategy for the growth of mesoporous TiO_2_ hollow spheres is illustrated in Figure 1a. After annealing and etching, the mesoporous TiO_2_ hollow spheres were obtained, as the shown in Figure 1b. From the TEM image (Figure 1c) of the single TiO_2_ hollow spheres, mesoporous structure and much tiny TiO_2_ nanocrystals are clearly found. Chueh et al. reported the synthesis of hierarchical NiCo_2_S_4_ hollow microspheres by using SiO_2_ spheres as hard template [74]. The SEM image indicates that the NiCo_2_S_4_ spheres has a hollow-structured interior (Figure 1d). The TEM image of the resulting hollow NiCo_2_S_4_ in Figure 1e reveals hollow spherical structures with thickness of ~60 nm. The magnified TEM image in Figure 1f shows flower-like microspheres comprising numerous intercrossed nanoflakes (~4 nm thick). And the TEM image reveals that the NiCo_2_S_4_ hollow spheres has a similar size and shape with the SEM image.

For soft-templating methods, the involved template (emulsion droplets, micelles, vesicles, microemulsion, and gas bubbles) are generally in the form of fluid/gas with high deformability. Thus, the complicated template elimination process is generally not necessary [75,76]. Typically, Wang et al. successfully synthesized the α-Fe_2_O_3_ hollow nanospheres through a facile quasiemulsion-templating approach [77]. In this synthesis, the reaction temperature influences the morphology of the α-Fe_2_O_3_. By adjusting the temperature of the reaction, the α-Fe_2_O_3_ hollow spheres with distinct packing densities of the nanosheets can be obtained as shown in Figure 1g–i. Wan et al. synthesized hollow and mesoporous Nb_2_O_5_ nanospheres (HM–Nb_2_O_5_) through a simple soft-templating method [78]. Urea play a bifunctional role in the synthesis. On the one hand, it acts as structural scaffold to form the nanospheric precursor. On the other hand, its gradual decomposition upon heating initiates the intraparticle transition of urea niobium oxalate into basic niobium oxalate. After water washing it forms hollow basic niobium oxalate, and then transformed into HM–Nb_2_O_5_ after heating at 600 °C (Figure 1j). Lee et al. synthesized the V_2_O_5_ microspheres (V2) by using the polyvinylpyrrolidone (PVP) aggregation as the soft template in the presence of ethylene glycol (EG) [79]. Figure 1k shows the hollow microsphere structure of V2 with an outer diameter of 2.7 μm. As shown in the inset of Figure 1l, the broken hollow sphere can confirm that these microspheres take a complete hollow structure.

Huang et al. synthesized the NiCo_2_O_4_ hollow microspheres through a simple template-free solvothermal method [80]. As shown in Figure 1m, the diameter of the NiCo_2_O_4_ hollow microspheres is approximately 2–3 μm. Due to the effectiveness of template-free method, that could also be extended to a one-pot process to compose several morphologies of the hollow spheres. For example, Wang et al. successfully synthesized the V_2_O_3_@C hollow spheres through a facile one-step solvothermal route [81]. The TEM image in Figure 1n shows that these V_2_O_3_@C hollow spheres consist of uniform microspheres with a diameter of approximately 1 μm, which is assembled with nanosheets. The hollow characteristic of the V_2_O_3_@C microspheres is confirmed by the magnified TEM image shown in Figure 1o, showing a shell thickness of approximately 85 nm.

For the synthetic methods of the hollow spheres, the hard template can be selectively etched, whereas the complicated template elimination process of soft template is generally not necessary. With this method, product uniformity is sometimes compromised. However, the possibility of producing more complicated hierarchical structures is largely increased by refilling a hollow interior with functional species or the in-situ encapsulation of guest molecules during shell formation. Therefore, the soft template method is more suitable for the preparation of hollow spheres.

### 2.2. Core-shelled Spheres with Complex Architectures

Core–shell nanostructures often possess superb chemical and physical properties compared with their single-component counterparts [82,83,84]. Hence, they are widely employed in optics, biomedicine, energy conversion, storage, etc. [85,86]. Core–shell structures can be broadly defined as a combination of a core (inner material) and a shell (outer layer material). Generally, many considerable efforts on core–shell materials have been reported, such as a solid inner core coated with one or more layers (shells) of different materials [87,88,89].

For the one solid inner core coated with one shell material, our group reported the preparation of highly uniform and shape-controlled Ni-CeO_2_@PANI (polyaniline) nanospheres [90]. Figure 2a illustrates the process for fabricating the core–shelled Ni-CeO_2_@PANI nanospheres (NCP1), yolk–shelled Ni-CeO_2_@PANI nanospheres (NCP2), and PANI hollow nanospheres. First, PVP aggregate is selected as capping agent and deposited on the surface of Ni-CeO_2_ nanospheres. The PANI shell was coated on Ni-CeO_2_ surface through the chemical oxidative in situ polymerization of aniline in hydrochloric acid (HCl) solution by using ammonium persulfate (APS) as oxidant. With the increasing of the HCl content, different morphologies nanocomposites were obtained, such as core–shelled Ni-CeO_2_ nanospheres, yolk–shelled Ni-CeO_2_ nanospheres, and PANI hollow spheres. Based on the TEM images, compared with Ni-CeO_2_ (Figure 2b), the diameter of core–shelled Ni-CeO_2_ nanospheres (Figure 2c) increased from 100 nm to 180 nm. Xu et al. synthesized the core–shelled TiO_2_@MoS_2_ microspheres through a hydrothermal method combined with annealing. Figure 2d shows the process used to fabricate the core–shelled TiO_2_@MoS_2_ microspheres [91]. Figure 2e displays the SEM image of the core–shelled TiO_2_@MoS_2_ microspheres after annealing. The MoS_2_ nanosheets grown evenly on the TiO_2_ surface and the average diameter of TiO_2_@MoS_2_ is approximately 580–620 nm. The TEM image displays the thickness of the MoS_2_ shells in the range of 130–170 nm (Figure 2f). Zhang et al. produced three-dimensional superstructures made of core–shelled SnO_2_@C nanospheres through a hydrothermal and sintering procedure [92]. The TEM image in Figure 2g reveals that the average diameter of the core–shelled SnO_2_@C nanospheres is approximately 50–60 nm, and the carbon thickness is approximately 10 nm. Shen et al. prepared the core–shelled SiO_2_@TiO_2_ microspheres by using carboxyl-modified SiO_2_ spheres as a core and an ethanol/acetonitrile mixture as solvent [93]. In this case, acetonitrile promotes the solubility and stability of titanium tetrabutoxide (TBOT) and restricts its hydrolysis, improving the control for the uniform deposition of TiO_2_ without the need of capping agents or special precursors. Figure 2h–j show the TEM images where the shell thickness can be facilely tuned from 12 nm to 100 nm by changing the TBOT concentration.

For the one solid inner core coated with two or more shells materials, Guo et al. reported the synthesis of the core–shelled Fe_3_O_4_/PANI/MnO_2_ hybrids [94]. Fe_3_O_4_ spheres were chosen as the inner core, follow coated by PANI and MnO_2_, respectively. Thereafter, core–shelled Fe_3_O_4_/PANI/MnO_2_ nanaospheres were formed. Figure 3a shows the typical TEM image of core–shelled Fe_3_O_4_/PANI/MnO_2_ composite, revealing uniform spherical nanostructures with a diameter of ~300 nm. The TEM image in Figure 3b confirms the uniform core–shell structure. The thickness of MnO_2_ nanoflakes is 5 nm, while that of the coating of MnO_2_ shells is approximately 50 nm. Our group reported core–shelled Fe_3_O_4_@C@MnO_2_ microspheres that were fabricated using multi-step solution-phase interface deposition [95]. Fe_3_O_4_ nanoparticles were coated with SiO_2_ through the Stöber method and further covered with resorcinol and formaldehyde (RF) resins. Fe_3_O_4_@C nanoparticles with inter-lamellar void were obtained by carbonizing RF under N_2_ atmosphere and etching SiO_2_ with NaOH. These nanoparticles served as a template, which were further coated with MnO_2_ shell to prepare Fe_3_O_4_@C@MnO_2_ microspheres. The TEM image in Figure 3c shows that the resultant composites have a typical core–shell structure with distinct magnetite core. The average diameter of the Fe_3_O_4_@C@MnO_2_ microspheres is ∼410 nm with 10 nm inter-lamellar void, a 30 nm thick carbon layer in the middle layer, and a 50 nm thick MnO_2_ shell in the outer layer. The SEM image of the as-prepared Fe_3_O_4_@C@MnO_2_ microspheres is shown in Figure 3d. Uniform flower-like MnO_2_ shells were formed and deposited onto the surface of the Fe_3_O_4_@C. The Fe_3_O_4_@C@MnO_2_ microspheres have an average diameter of ∼410 nm, which is consistent with their corresponding TEM findings. Zhu et al. reported core–shelled SiO*_x_*-TiO_2_@C nanocomposites synthesized through a scalable sol-gel method combined with carbon-coating [96]. Figure 3e shows the field-emission SEM (FESEM) image of the core–shelled SiO*_x_*–TiO_2_@C nanospheres with an average diameter of 100 nm. In Figure 3f, the TEM image shows the presence of outer turbostratic carbon shell thickness of ~8 nm, in which the inner cores SiO*_x_*–TiO_2_ were fully coated.

The main advantages of these core–shell structures include the following ability to: (1) protect the core from the effect of environmental changes outside; (2) intensify or introduce new chemical or physical capabilities; (3) limit volume expansion and maintain structural integrity; (4) protect the core from aggregating into large particles; and (5) percolate ions or molecules onto the core selectively.

### 2.3. Yolk-shelled Spheres with Complex Architectures

Deviating from the core–shelled structure, a typical yolk–shelled spherical structure has a smooth shell and core which can be also called yolk, and the shell with core has a void space, which provides movable space for the inter yolk. Both their shells and yolks generally have variations, such as a single shell with a single yolk [97], double shells with a single yolk (yolk–shells) [98,99], multi-shells with a single yolk (yolk–shells) [100,101], and a single shell with multi-yolks (yolks–shell) [102]. Yolk–shelled structure materials were first synthesized through silica template by Hyeon et al. Initial researches of yolk–shelled structures concentrated on spherical structures. For a better understanding of these structures, we will review these materials based on different structure types, as shown in Figure 4 and Figure 5.

For the single shell with single yolk spherical structure, Pan et al. reported the synthesis of yolk–shelled MoS_2_@C microspheres through a solvothermal method combined with annealing [103]. First, the MoS_2_ microspheres were synthesized through a solvothermal method in EG solution. The obtained MoS_2_ microspheres were coated with polydopamine (PDA) to form core–shelled MoS_2_@PDA microspheres, which were carbonized to yield core–shelled MoS_2_@C microspheres (MoS_2_@C-0) through annealing. Subsequently, hydrogen peroxide (H_2_O_2_) solution was used to etch the MoS_2_ microspheres. Different concentrations of H_2_O_2_ solution (0.2, 0.4, and 0.6 vol%) were used, resulting in the formation of different sizes of the void space in the MoS_2_ yolk and the carbon shell, which were denoted as MoS_2_@C-0.2%, MoS_2_@C-0.4%, and MoS_2_@C-0.6%, respectively (Figure 4a). MoS_2_@C-0.4% composed of a 25 nm thick porous carbon layer, a 280 nm nanosized MoS_2_ yolk, and well-controlled internal void in between (Figure 4b). Wen et al. reported the preparation of yolk–shelled SiO_2_@C nanospheres through two steps [104]. First, the synthesis of SiO_2_ nanospheres through Stöber’s method. Second, coating the RF polymer layer on SiO_2_ nanospheres surface and follow carbonization. The SiO_2_ nanospheres inside core–shelled SiO_2_@C were controllably etched through a hydrothermal method that produces yolk–shelled SiO_2_@C nanospheres. During this progress, SiO_2_ is transformed into Si(OH)_4_, which is dissolved under high presure and temperature [105]. Through this method, the size of the void between the SiO_2_ yolk and carbon shell could be efficiently controled by adjusting the etching time, temperature, and solution concentration. Typical TEM images of the yolk–shelled SiO_2_@C (Figure 4c,d) reveal that the thickness of carbon shell is approximately 10 nm. Zhao et al. synthesized the yolk–shelled Fe_3_O_4_@RF@void@mSiO_2_ nanospheres through the swelling–shrinkage of RF upon soaking in or the removal of organic solvent [106], which has a Fe_3_O_4_@ RF core and a mSiO_2_ shell. In Figure 4e, TEM image shows that the obtained Fe_3_O_4_@RF@void@mSiO_2_ nanospheres possess uniform and well-dispersed spherical morphology with a diameter of 472−638 nm. The magnified TEM image in Figure 4f can be clearly found the inner RF-protected magnetic Fe_3_O_4_ core.

Recently, many novel yolk–shelled spheres have emerged. Different from the typical yolk–shelled spheres with smooth surface of shell and yolk, their shell or yolk possesses various surface structures. Below, we introduce two interesting works with coconut-like and flower-like yolk–shelled spheres, respectively, using surfactant aggregation as templates. Coconut-like yolk–shelled PS@NiCo_2_S_4_ nanosphere (Figure 4g) was synthesized from interior to exterior by Zhu et al. [107]. SiO_2_ nanospheres was used for the hard template and then removed during the hydrothermal process. Figure 4h exhibits the nanosphere with a PS yolk and a numerous NiCo_2_S_4_ nanosheets around the shell. Our group successfully synthesized the flower-like yolk–shelled SiO_2_ nanospheres (FYSSns) through a facile one-pot strategy by using CTAB–PVP composite surfactant aggregation as soft template and cyclohexane–ethanol–water as microemulsion [108]. When added TEOS, with the hydrolysis and condensation of TEOS, SiO_2_ shells and flower–shaped yolks were formed in the hydrophilic region of the composite templates, respectively. After calcination, microemulsion aggregations and vesicles were removed and the FYSSns were obtained, as shown in Figure 4i. The TEM images in Figure 4j–l show that these FYSSns were evenly dispersed with an average diameter ranging of 500–600 nm, and a large space between the shells and the flower-shaped yolks were observed. The flower-shaped yolks diameter ranged of 260–320 nm, the space of yolk and shell ranged of 100–120 nm, and the shell thickness ranged of 20–30 nm.

Aside these surfactant aggregation templates, organic solvents could also generate the soft templates in oil/water systems. For instance, Peng et al. successfully synthesized the yolk–shelled CoS_2_ nanospheres with various interior composition through a facile solution-based route [109]. The concentration of Carbon disulfide (CS_2_) oil droplets has an important effect on the morphology of the product. When added 0.4 mL of CS_2_, the product shown in Figure 4m is yolk-shelled CoS_2_ nanospheres with an average diameter of 800 nm. From an individual broken sphere, the interior yolk with a diameter of about 300 nm, a void between the yolk and the shell is about 100–200 nm, and both of the yolk and shell are constructed by nanosheets (Figure 4n).

Apart from a single shell with single yolk, yolk–shells, and yolks–shell structure were reported, which are shown in Figure 5. For example, Figure 5a shows that double-shelled SnO_2_ yolk–shells nanospheres was fabricated by Hong et al. through carbon calcination for three times [98]. In this yolk–shelled structure design, carbon was used as the hard template. The carbon was formed inside the SnO_2_ nanospheres by the polymerization and carbonization of sucrose, resulting in the precursor of C–SnO_2_ nanospheres. The first combustion fabricated the core-shelled C–SnO_2_/SnO_2_ nanospheres. The second step of combustion of the core-shelled C–SnO_2_/SnO_2_ nanospheres was supplied to form the yolk-shelled C-SnO_2_@SnO_2_ nanospheres. Further heating the core-shelled C–SnO_2_@SnO_2_ nanospheres, a yolk with double-shells structure SnO_2_@SnO_2_@SnO_2_ was generated. Leng et al. successfully used PVP as the surfactant and template to synthesize the triple-shelled NiCo_2_O_4_ yolk–shells nanospheres by spray pyrolysis [100]. Figure 5b shows that the triple-shelled NiCo_2_O_4_ yolk–shells nanospheres had uniform sizes. TEM image in Figure 5c shows that the products prepared with PVP exhibits three shells with one yolk. As shown in Figure 5e, the elemental mapping images further show that the Ni, Co, and O are evenly distributed in the triple-shelled NiCo_2_O_4_ yolk–shells nanospheres. In addition to the single yolk with multi-shells, a single shell with multi-yolks Sn_4_P_3_@C nanospheres were fabricated [102]. The TEM image (Figure 5d) shows that the chief yolk of the Sn_4_P_3_@C composite materials was abounded with much tiny yolks. Our group reported a new method to prepare the multi-yolks with single shell SiO_2_–TiO_2_ (pomegranate-like) microspheres through a three-step approach [110]. First, SiO_2_–hydrophobic poly(methyl methacrylate) (PMMA)–hydrophilic poly(oligo(ethylene glycol)methyl ethermethacrylate) (POEOMA) microspheres were fabricated through aqueous polymerization. After coating the TiO_2_ shells on the modified SiO_2_ spheres surface and removing the PMMA–POEOMA polymerlayer through calcination, the multiple yolks with single shell SiO_2_@TiO_2_ microspheres were obtained. Figure 5f illustrates the process used to fabricate the multiple yolks with single shell SiO_2_@TiO_2_ microspheres through the hydrolysis and condensation of TBT. Figure 5g shows the TEM image of the multiple yolks with single shell SiO_2_@TiO_2_ microspheres after calcination. Yolks–shell structured SiO_2_@TiO_2_ microspheres have a smooth surface with the diameter of approximately 55 nm. The FESEM image (Figure 5h) reveals that each SiO_2_@TiO_2_ microsphere is composed of multiple SiO_2_ yolks and a single TiO_2_ shell. This structure shows the typical multiple yolks with single shell structure.

Typical spherical yolk–shelled structures are tuned with various numbers of shells and yolks. The suitable void space between yolk and shell can accommodate the volume expansion of yolk and avoid aggregation of electroactive cores during charging/discharging process. With the development of different synthetic methods, yolk–shelled structures can be prepared into manifold types.

### 2.4. Double-shelled Spheres with Complex Architectures

Typically, double-shell micro/nanostructured spherical materials often possess double shells, hollow core, and a gap or no gap between the double shells, which are shown in Figure 6. The combination of layer-by-layer (LBL) coating with a selective etching procedure is often utilized to prepare the double-shelled or multi-shelled hollow structures [111]. As a specific example, Li et al. first reported the preparation of the anatase–rutile TiO_2_ double-shelled hollow spheres (DSHSs) through a facile sol-gel method using SiO_2_ nanospheres as the hard template [112]. TiO_2_ and SiO_2_ shells grown alternately on the inner SiO_2_ cores into onion-like SiO_2_@TiO_2_@SiO_2_@TiO_2_ nanospheres. The initial SiO_2_ core and the SiO_2_ layer between the two TiO_2_ shells work as the hard template. After annealing and etching, anatase–rutile TiO_2_ DSHSs were obtained (Figure 6a). The TEM image in Figure 6b clearly shows that the diameter of the inner core is approximately 210 nm. The thicknesses of the TiO_2_ DSHSs outer shell is approximately 30 nm, and the inner shell is approximately 35 nm. Theoretically, LBL is a powerful method but time-consuming. In the same way, the preparation of the SnO_2_@C DSHSs have been reported by Lou et al. [113]. In this synthesis, SiO_2_ nanospheres were used as the hard template and successively coated with SnO_2_ double-shells, and then coated with glucose-derived carbon-rich polysaccharide (GCP) layers on these core-shelled SiO_2_@SnO_2_ nanospheres through hydrothermal processes. After carbonization and SiO_2_ removal, the SnO_2_@C DSHSs were finally obtained. The DSHSs structure of the SnO_2_@C can be clearly obtained in Figure 6c,d. At the same time, Yang et al. successfully used a hard-templating method to prepare the SnO_2_@C DSHSs [114]. In this strategy, the SnO_2_ LBL coating, GCP layer coating, and SiO_2_ inner core etching are avoided. Instead, the three processes are achieved simultaneously. Lou et al. developed the double-shelled hollow carbon spheres (DHCSs) through the hard templates method [115]. SnO_2_ hollow nanospheres synthesized through the solvothermal method were selected as the hard templates for depositing the GCP layer on both inner and outer surface. After annealing in H_2_/N_2_ atmosphere, the SnO_2_ core was then dissolved using an acid to generate the DHCSs. In the magnified TEM image (Figure 6e), the double-shelled structure can be easily recognized.

Our group reported the preparation of p–n heterostructured TiO_2_/NiO DSHSs nanomaterials [116]. SiO_2_ nanospheres (the diameter is ∼270 nm) were selected as hard template and further coated with TiO_2_ layer through sol–gel method. After annealing and etching, the TiO_2_ hollow spheres were obtained. Then, NiO layer was coated onto the TiO_2_ hollow spheres surface through the hydrothermal process. The p–n heterostructured TiO_2_/NiO DSHS was obtained after the annealing process. Figure 6f–i shows that the shell thickness of NiO increases with the weight of NiO. By using a similar coating method, Qian et al. prepared N-doped DHCSs (N-DHCSs) by selecting Fe_3_O_4_ porous hollow nanospheres as the hard templates [1]. Manthiram et al. fabricated the N-DHCSs by using TiO_2_ hollow nanospheres as the hard templates and dopamine as the N-doping carbon precursor [117]. Recently, our group also successfully synthesized CeO_2_@RF DSHSs through the polymerization of resorcinol and formaldehyde on the surface of CeO_2_ hollow nanospheres [118]. After carbonizing, DSHSs mesoporous CeO_2_@C were obtained. In Figure 6j, the TEM image of CeO_2_@C displays a typical DSHS structure with a cavity diameter of approximately 60 nm. The CeO_2_ shell of the interior layer of the CeO_2_@C with a thickness is approximately 20 nm, while that of the carbon shell of the outer layer with a thickness is approximately 15 nm. 

Aside from SiO_2_- and carbon-based materials, surfactant aggregate can also be used as the soft template for the preparation of DSHSs structure with other materials complex such as TMOs. Wang et al. synthesized the single-crystalline Cu_2_O DSHSs (Figure 6k) by using CTAB vesicle as the soft template through adjustment the concentration of CTAB in aqueous solution [119]. Liu et al. developed a simple template-free solvothermal route and subsequent heating treatment process for the synthesis of the V_2_O_5_−SnO_2_ DSHSs [120]. Figure 6l and m show the TEM images of the typical V_2_O_5_−SnO_2_ DSHSs structure. In Figure 6m, it can be clearly seen that the diameter of the inner cavity size is approximately 250 nm, and the thickness of the inner and outer shells is approximately 90 nm.

In addition, novel synthetic strategy has been reported for the synthesis of DSHSs, such as ion exchange approach. Ion exchange include either cations exchange and anions exchange between a solution and an insoluble solid. Recently, ion exchange approach has been developed as an effective method for the preparation of the DSHSs. Lou et al. successfully extended the ion exchange approaches for the synthesis of the NiCo_2_S_4_ DSHSs [121]. In this work, NiCo–glycerate spheres precursor were synthesized through a facile solvothermal method, and then a solution sulfidation process was utilized to convert the precursor into NiCo_2_S_4_ DSHSs. As shown in Figure 6n, the sulfidation progress mainly included three stages. At stage I, the core–shelled NiCo-glycerate@NiCo_2_S_4_ nanospheres were obtained from S^2−^ ions reacting with NiCo–glycerate spheres precursor at a high temperature. At stage II, the void gap between the NiCo-glycerate inner core and NiCo_2_S_4_ shell was produced due to the slow inward diffused S^2−^ ions and the fast outward diffused M^2+^, while the second shell of NiCo_2_S_4_ was formed. After the reaction of the anion (S^2−^) ions exchange reaction, the NiCo_2_S_4_ DSHSs were obtained at the end of stage III. Figure 6o,p show the NiCo_2_S_4_ DSHSs with rough surface and the diameter is around 250 nm.

For the double-shell micro/nanostructured spherical materials, more synthetic methods such as LBL growth, sol-gel method, hard template method, soft template method, and ion exchange approaches have been reported. LBL growth and hard template method are simple, effective, and straightforward in concept, whereas the soft template method generally be free from the complicated template elimination process. A suitable void space in inner or between the double shells can accommodate the volume expansion, increase the specific surface area of the material, and increase the contact area between the material and the electrolyte. In order to further increase the specific surface area of the material, the mesoporous features could be introduced to double-shell structures.

### 2.5. Multi-shelled Spheres with Complex Architectures

Multi-shelled hollow spheres (MSHSs) include multi-shells and a hollow chamber. Despite the challenges brought by the high structural complexity of these MSHSs, similar to single-shelled ones, they can also be synthesized based on well-controlled templating or self-templated methods. In this section, the synthetic methodologies for MSHSs, including well-established hard-, soft-, and self-templating methods, will be reviewed. 

#### 2.5.1. Hard-Templating Method

Hard-templating method is facile and straightforward theoretically, which is firstly used to prepare MSHSs structures. In general, the hard-templating synthesis involves four major steps: i) template preparation; ii) surface modification of the hard template, iii) target material coating/deposition; and iv) template removal. The coating/deposition of the target material on a hard template is generally considered as the most challenging step. Sometimes, the surface modification can be omitted if the target material is compatible with the template. The most frequently employed hard templates include monodisperse polymer, silica, carbon, metal, and metal oxide colloids. 

LBL growth is a typical hard-templating method with the repetition of coating procedure and selective etching to generate the formation of the MSHSs [122,123]. For example, Jang et al. synthesized the TiO_2_ MSHSs through LBL methods [124]. TiO_2_ and SiO_2_ shells are alternately grown on the surface of the inner SiO_2_ cores to form onion-like SiO_2_@TiO_2_@SiO_2_@TiO_2_@SiO_2_@TiO_2_ nanospheres. After annealing and etching, the TiO_2_ triple-shelled hollow spheres (TSHSs) can be obtained. Carbonaceous microspheres (CMSs) were usually synthesized by glucose through hydrothermal, and then dispersed into the 0.05–0.5 mol L^−1^ metal–salt solutions. After calcination, the metal oxide MSHSs are obtained [125,126,127,128]. For example, the Co_3_O_4_ MSHSs were obtained by using CMSs templates [129]. The morphology of the products was related to the different kinds of solvent. When the solvent was water only, CMSs dispersed into 1.0 mol L^−1^ [Co(H_2_O)_6_]^2+^, only Co_3_O_4_ single-shelled hollow spheres (SSHSs) (Figure 7a) were obtained after calcination. When the solvent was the mixture of ethanol and water (*v*/*v*=1:1), after calcination, the Co_3_O_4_ DSHSs (Figure 7b) were obtained. At a higher temperature, [Co(H_2_O)_6−*x*_]^2+^ can be easily adsorbed by CMSs, thus, the Co_3_O_4_ TSHSs are generated (Figure 7c). Treatment of the CMSs with HCl results in the formation of Co_3_O_4_ quadruple-shelled hollow spheres (QSHSs) (Figure 7d). Similarly, Cao et al. synthesized the nanorod-assembled Co_3_O_4_ MSHSs through a novel strategy [66]. Firstly, Co_2_CO_3_(OH)_2_ nanorods are vertically grown on CMSs surface to form the core-shelled CS@Co_2_CO_3_(OH)_2_ spheres through a low-temperature solution reaction. After annealing the CS@Co_2_CO_3_(OH)_2_ precursor in air, the Co_3_O_4_ MSHSs were unconventionally obtained. He et al. successfully synthesized the NiCo_2_O_4_ SSHSs, DSHSs, and TSHSs by controlling the penetrated amount of Ni^2+^ and Co^2+^ as shown in Figure 7e [130]. The TEM images of the NiCo_2_O_4_ hollow microspheres demonstrate the relatively uniform spherical morphologies with a diameter of ~200–300 nm. When the solvent was deionized water, the NiCo_2_O_4_ SSHSs (Figure 7f) were prepared with a heating ramp rate of 2 °C min^−1^. When the solvent was changed to EG, NiCo_2_O_4_ DSHSs (Figure 7g) and TSHSs (Figure 7h) were obtained with a heating ramp rate of 2 and 5 °C min^−1^, respectively. Xi et al. also prepared the WO_3_ QSHSs (Figure 7i) by using CMSs as hard template [131]. Zhou et al. directly prepared the α-Fe_2_O_3_ QSHSs (Figure 7j) through CMSs as hard template after calcinations [132].

#### 2.5.2. Soft-Templating Method

Soft-templating methods for the MSHSs structures usually use surfactant aggregates as a template such as supramolecular micelles, composite surfactant aggregates, and polymer vesicles [76]. However, the MSHSs structure is easily influenced by pH, temperature, solvent, and ionic strength [39]. Therefore, only a few successful examples are obtained, which are shown in Figure 8.

First, some successful examples are reported by the author to prove that we have a full understanding of this synthesis method [133,134]. In 2007, the highly regular SiO_2_ MSHSs (vesicle-like) were prepared by using P123 as surfactant and 1,3,5-triisopropylbenzene (TIPB) as hydrophobic additive through the hydrolysis and condensation of TEOS [135]. Figure 8a,b show the HRTEM images of the SiO_2_ MSHSs with the alternating concentric SiO_2_ shells and voids, and the shell thickness is about 5 nm. Figure 8c shows the typical FESEM image of the SiO_2_ MSHSs with spherical structure and rough surface. In 2012, the SiO_2_ MSHSs was successfully synthesized by utilizing the mixed surfactants (cationic surfactant CTAB and anionic surfactant sodium dodecyl sulfate (SDS)) aggregates as the soft template and TIPB as the micelle expander [136]. Figure 8d shows the interlamellar void between the SiO_2_ shells is approximately 15–20 nm, and the shell thickness is approximately 5–15 nm. In 2014, the SiO_2_ MSHSs was synthesized by using the didodecyldimethylammonium bromide (DDAB)/CTAB as co-surfactant template through the hydrolysis, condensation of TEOS and annealing [137]. Figure 8e shows the mechanism of the formation of multilayer vesicles. The molar ratio of DDAB and CTAB plays a major role in inducing the numbers of layers of the SiO_2_ MSHSs. With an increased amount of DDAB, the number of layers of the SiO_2_ MSHSs were effectively increased. Therefore, when the molar ratio of CTAB and DDAB was performed at 1:0.832, the SiO_2_ hollow nanospheres with few shells was formed. As the molar ratio of CTAB and DDAB continuously increased to 1:1.104, the SiO_2_ MSHSs with 6–7 layers was formed. As shown in Figure 8f,g, the diameter of the SiO_2_ MSHSs is 80–90 nm, the shell thicknesses is 2–6 nm, and the void gap is 2–3 nm. Moreover, this SiO_2_ MSHSs can be used as a hard template to synthesize other multilamellar vesicular materials. In 2017, as a typical example, the PANI MSHSs was firstly fabricated through a facile two-step method by using the SiO_2_ MSHSs as hard templates [138]. The PANI@RGO MSHSs composites were prepared by self-assembling GO onto the PANI MSHSs surface and followed by hydrothermal reduction process. The TEM image in Figure 8h shows that the SiO_2_ MSHSs was successfully removed to obtain the PANI MSHSs. 

Chen et al. successfully synthesized the CoFe_2_O_4_ MSHSs with a tunable number of 1–4 layers through a facile one-step method by using cyclodextrin as a surfactant template, followed by calcinations [139]. The shell number and porosity of the CoFe_2_O_4_ MSHSs can be controlled by adjusting the synthesis parameters. Figure 8i shows the TEM image of the CoFe_2_O_4_ QSHSs. This is the first report using cyclodextrin as template for accurate synthesis of shell-controllable CoFe_2_O_4_ QSHSs. Wang et al. reported the preparation of the Cu_2_O MSHSs by using cationic surfactant CTAB as the soft template [119]. By adjusting the concentration of CTAB from 0.10 mol L^−1^ to 0.15 mol L^−1^, the structure of Cu_2_O can be tuned from SSHSs (Figure 8j), DSHSs (Figure 8k), TSHSs (Figure 8l), to QSHSs (Figure 8m). Similarly, Liu et al. synthesized the mesoporous SiO_2_ MSHSs with a controllable shell number of 1–4 layers [140]. Gu et al. also successfully synthesized the mesoporous carbon MSHSs with a controllable shell number of 3–9 layers [141], further demonstrating the feasibility of producing the MSHSs structures through soft-templating method.

#### 2.5.3. Self-Templating Method

As an emerging approach, the self-templating method is different from the traditional hard-/soft-templating methods. The precursors of self-templating method composites are not only used as the self templates to form the MSHSs structures but also transformed into the fundamental compositions of the ultimate products [142,143,144]. This strategy does not require the removal of template, which simplifies the synthetic processes and decreases the production costs [145]. Several synthesis mechanisms of MSHSs structures have been reported, such as Ostwald ripening approach [146,147], Kirkendall growth [148,149], and galvanic exchange [150,151].

Several typical and related efforts on the preparation of the novel MSHSs structures have been reported. For example, Tang et al. reported the preparation of the CeO_2_ TSHSs through a self-templating method, which are composed of much tiny CeO_2_ nanoparticles [152]. Firstly, the carbon microspheres were formed through glucose in aqueous solution. Subsequently, the Ce^3+^ ions are adsorbed onto the carbon microspheres in an alkaline environment through electrostatic attractions. Finally, after calcination in air, the TSHSs CeO_2_ were obtained. The TEM image demonstrates that CeO_2_ have a relatively uniform spherical structures with a diameter of ~1–2 μm (Figure 9a), while the TEM image shows that these CeO_2_ microspheres are exclusively characteristic TSHSs structures (Figure 9b). A programmed temperature strategy was proposed by Xie et al. to synthesize the MSHSs from the solid templates [151]. Solid particles initially changed to core–shell structures and then changed to the completely SSHSs structures due to the Kirkendall effect. The second shell can be prepared by decreasing the reaction temperature to inhibit the formation of the hollow structures and then increasing the temperature again. By repeating the process for several times, the MSHSs structures are expected to be achieved.

Recently, the inside-out Ostwald ripening approach has been further demonstrated to be an efficient procedure in synthesizing the MSHSs structures. For instance, the Cu_2_O MSHSs were prepared by Zhang and Wang through a multistep Ostwald ripening processes [153]. From the first Ostwald ripening approach, the Cu_2_O SSHSs were obtained (Figure 9c). With the introduction of extra reactants into the reaction mixture, new Cu_2_O nanoparticles were produced and deposited on the surface of the first shell. When the second Ostwald ripening approach occurred, both the inner and the outer Cu_2_O shells would become thinner gradually, and as shown in Figure 9d, the Cu_2_O DSHSs were obtained. By repeating the process for three and four times, the Cu_2_O TSHSs (Figure 9e) and QSHSs (Figure 9f) can also be easily prepared, respectively.

Table 1 generalizes some typical instances about the MSHSs structures, including their composition, morphology, and synthetic method. Notably, some novel preparation methods have been developed to synthesize the MSHSs structures, but these methods are not included in the above-mentioned three types. For example, Zeng et al. reported an ion exchange approach to prepare the Cu_2_S SSHSs, DSHSs, TSHSs, and QSHSs [154]. Lu et al. designed periodic mesoporous organosilica MSHSs through the selective etching of “soft@hard” particles [155]. Lastly, González et al. developed multi-metal hollow nanoparticles with complex morphologies and composition, such as concentric double-shelled nanoboxes, through sequential galvanic exchange and Kirkendall effect [156].

Hard templates are monodisperse, easy size and shape controllable, ready availability in large amounts, and easy synthesis using well-known recipes. In spite of these advances, hard-templating methods still face quite a few challenges, such as the difficulty to achieve uniform coating due to compatibility issues between the templates and desired shell materials, and the tedious template removal processes. However, the MSHSs structure is easily influenced by pH, temperature, solvent, and ionic strength. Unlike conventional hard-/soft-templating approaches, self-templating synthesis enables good control over the particle uniformity without an auxiliary template removal process, which simplifies synthetic procedures, reduces production costs, and provides ease for scale up. Hence, modified templating strategies with an extra conversion step could transfer compatible precursor shells into target materials. Through these synthetic protocols, core and shell substances are elaborately selected to avoid potential compatibility problems.

## 3. Applications of Micro/nanostructured Spherical Materials in Energy Storage

### 3.1. Lithium-Ion Batteries

LIBs have been widely used in various energy storage devices, portable electronic devices, static storage media, EVs, and hybrid EVs due to their beneficial characteristics, such as high specific capacity and energy density, and long cycle life [96,157,158]. The electrochemical performance of LIB is mainly determined by the electrode materials [159,160]. In order to seek for advanced anode materials, various high-capacity candidates have been researched, such as transition metal oxides and sulfides [161,162,163,164,165]. Nevertheless, their application has been hampered by the rapid fade in specific capacity, which is related with the large volume change of the electrode materials during Li^+^ intercalation/deintercalation process and the poor rate capability due to the low diffusion rate of Li-ions.

The design and fabrication of electrode materials possessing specific nanostructures is advantageous in solving the above problems. Motivated by the versatility of spherical structures, various morphology of electrode materials with improved electrochemical performance, including hollow, core-shelled, yolk-shelled, double-shelled, and multi-shelled spheres have been fabricated. For example, Wang et al. reported the formation of porous V_2_O_3_@C hollow spheres composed of ultrathin nanosheets (Figure 10a) [81]. Comparing with several vanadium based hollow materials, V_2_O_3_@C hollow spheres exhibit high reversible capacities as well as superior rate performance when they were used as anode materials for LIBs. Figure 10b shows the galvanostatic charge and discharge profiles of the V_2_O_3_@C hollow spheres electrode material at the different current densities. As for the long cycling performance of this electrode material, a discharge specific capacity of 583 mA h g^−1^ can be maintained after 800 cycles at a high current density of 2 A g^−1^, and that can be clearly seen from Figure 10c, the capacity retention of over 100%. The increased capacity might well be attributed to the hollow spheres structure that could approve the penetration of electrolyte solution and also buffer volume change of electrode material during the lithiation/delithiation processes [166]. Yang et al. reported the SnO_2_@C DSHSs prepared by using SiO_2_ sphere as hard templates [114]. Figure 10d distinctly displays that SnO_2_@C consists of peculiar nanostructure of the DSHSs. Figure 10e reveals the rate capability of the SnO_2_@C DSHSs at the different current densities from 400 mA g^−1^ to 3000 mA g^−1^. Figure 10f shows that the SnO_2_@C DSHSs exhibits a superior cycling stability, delivering a high reversible capacity of 838.2 mA h g^−1^ at the current density of 200 mA g^−1^ even after 500 cycles. In contrast, the SnO_2_ hollow spheres (SnO_2_ HS) shows the poor cycling performance and fades drastically. It can be confirmed that the double-shell structure has excellent structural stability. Our group reported a TiO_2_/C/MoS_2_ composite through solvent thermal method [5]. When used as the anode materials for LIBs, compared with the pure TiO_2_ or MoS_2_, TiO_2_/C/MoS_2_ microspheres can significantly enhance the electrochemical performance, showing a high initial discharge specific capacity of 1219 mA h g^−1^, and after 100 cycles, 621 mA h g^−1^ remained at 100 mA g^−1^. Our group also reported the mesoporous CeO_2_ DSHSs, and their TEM image is shown in Figure 10g [118]. Figure 10h shows the charge and discharge voltage profiles of the CeO_2_@C DSHSs at the current density of 100 mA g^−1^. In the first cycle, CeO_2_@C reveals a high initial charge and discharge specific capacities of 781.6 and 1309.1 mA h g^−1^, respectively. In terms of the cycling performance of the CeO_2_@C DSHSs, as shown in Figure 10i, a high discharge specific capacity of 903.6 mA h g^−1^ remained at a current density of 100 mA g^−1^ even after 300 cycles. And the Coulombic efficiency (CE) of CeO_2_@C DSHSs is approximately 98.7%, indicating that CeO_2_@C DSHSs has a high specific capacity and a good cycle performance.

Although they possess above mentioned advances, simple hollow nanostructures can only provide limited possibilities to modulate the properties of electrode materials. Therefore, further manipulation of hollow structures in terms of geometric morphology, chemical composition, and shell architecture for complex architectures is required to achieve improved electrochemical performance demanded by the current emerging technologies. Lu et al. reported a novel design and preparation strategy for the synthesis of void-controlled yolk–shelled MoS_2_@C-0.2% (Figure 11a), MoS_2_@C-0.4% (Figure 11b), and MoS_2_@C-0.6% (Figure 11c) microspheres, respectively [103]. For the yolk–shelled MoS_2_@C-0.4% microsphere anode, the initial discharge specific capacity reached 1813 mA h g^−1^, a high reversible capacity of 1016 mA h g^−1^ after 200 cycles (Figure 11d), and superior rate capability. Choi and Kang prepared the yolk-shelled NiO nanospheres through the continuous one-pot of spray pyrolysis [167]. The diameter of core and the thickness of shell were about 200 and 75 nm, respectively. When used as the anode material, the yolk-shelled NiO delivered initial discharge capacity of 1200 mA h g^−1^ at the current density of 700 mA g^−1^. After 50 cycles, the discharge specific capacity of the yolk-shelled NiO was as high as 824 mA h g^−1^ at a high current density of 1 A g^−1^.

The MSHSs structure can extend the long cycle performance of electrode materials because of the enhanced structural stability, and the shell numbers could affect the properties of Li storage [129,132]. In a related work, Wang et al. analyzed the effect of shell number on the performance of Li storage in detail [129]. The initial capacities of Co_3_O_4_ SSHSs, DSHSs, TSHSs, and QSHSs are 1087.2, 1450.0, 2063.7, and 1626.2 mA h g^−1^, respectively, all of which are larger than the theoretical capacity of Co_3_O_4_ (890 mA h g^−1^). As shown in Figure 11e, the α-Fe_2_O_3_ QSHSs have been prepared through CMSs as hard template after calcinations in air [132]. The resulting electrode material of α-Fe_2_O_3_ QSHSs shows high discharge specific capacity, excellent long cycle performance, and superior rate capability. At a current density of 50 mA g^−1^, α-Fe_2_O_3_ QSHSs deliver an initial discharge and charge specific capacities of 1443 mA h g^−1^ and 1067 mA h g^−1^, respectively (Figure 11f). After 50 cycles, high and stable specific capacities of ~1000 and 900 mA h g^−1^ can be obtained at the current densities of 400 and 1600 mA g^−1^, respectively (Figure 11g). As shown in Figure 11h, when the current densities increase from 100 to 3200 mA g^−1^, the capacities decrease slightly from 1228 to 784 mA h g^−1^. With the current density decreases to 200 mA g^−1^ again, the capacity recovers to 1176 mA h g^−1^, indicating the excellent rate capability of the α-Fe_2_O_3_ QSHSs. Its capacity is stable and its cycle stability is obviously improved. This is an effective strategy for promoting the electrochemical performance of LIBs. Yao et al. successfully synthesized the Co_3_O_4_ MSHSs through a polyol process which can be employed as anode material for LIBs. The effect of shell numbers on Li storage performance was analyzed in this work [168]. They first prepared the Co_3_O_4_ SSHSs, DSHSs, and TSHSs through a PVP-mediated solvothermal method. The TEM image of the Co_3_O_4_ TSHSs is shown in Figure 11i. As shown in Figure 11j, the first cycle discharge and charge curves of the samples show that the Co_3_O_4_ TSHSs have the highest discharge specific capacity. After 50 cycles, the capacities of the Co_3_O_4_ SSHSs, DSHSs, and TSHSs remain at 680, 866, and 611 mA h g^−1^, respectively (Figure 11k). Figure 11l shows the rate capability of the Co_3_O_4_ DSHSs, a capacity of 500.8 mA h g^−1^ even at a high current density of 2 C, exhibiting a good rate capability. Lou et al. further certificated the advantages of the MSHSs structures as anode electrode materials for LIBs, including the CoMn_2_O_4_ MSHSs structures [169], the Fe_2_O_3_ multi-shelled microcages [170], and so on.

Improved lithium storage performance has been realized by simultaneously manipulating the morphology of hollow particles and their compositions. Rational incorporation of multi-compositions in the shells of hollow nanostructures might combine the advantages of different materials, and the synergistic effect arising from their interaction promises improvement for lithium storage performance. The multi-shelled hollow structures can improve the volumetric energy density of electrodes by increasing the weight fraction of the active species, and also extend the cycle life due to the enhanced structural stability.

### 3.2. Lithium–Sulfur Batteries

Recently, LSBs have been regarded as prospective candidate of the energy storage device due to their high theoretical specific capacity and energy density, abundance of sulfur, low cost, and environmental friendliness [171,172,173]. However, the intrinsic challenges of LSBs have limited their practical applications: the poor electrical conductivity of S and Li_2_S results in low sulfur utilization, the volume expansion from S to Li_2_S during the lithiation/delithiation processes, and the “shuttle effect” induced by the diffusion of polysulfide intermediates (Li_2_S*_n_*, 3≤*n*≤8) [174,175,176,177,178]. Therefore, the preparation of the suitable S hosts which can promote the charge transport and limit the produce of polysulfides is essential for boosting the performance of LSBs [179,180,181,182,183].

To suppress the dissolution of polysulfides and maintain a high S utilization of LSBs, Li et al. successfully synthesized the SiO_2_@TiO_2_ DSHSs with radial meso-channels [184]. As shown in Figure 12a, the DSHSs hollow structures can also be clearly identified, and the thickness of the TiO_2_ shell is around 10 nm. From the HRTEM image in Figure 12b, the meso-channels throughout the outer TiO_2_ layers can be observed, suggesting that even the core part of the composite material can readily contact with the electrolyte. Due to the unique structures and compositional advantages, a better capacity retention is achieved to 65.5% over 500 cycles at 0.5 C (Figure 12c). In comparison, a high S of up to 80 wt% is achieved with about 33% capacity retentions over 1000 cycles at 1 C, as exhibited in Figure 12d. Lou et al. developed the DHCSs by using SnO_2_ hollow sphere as hard templates [71]. The double-shelled structure can be easily recognized from TEM image in Figure 12e. Figure 12f,g show the discharge–charge cycling performance at 0.8 C and the rate capability after 100 cycles of the DHCS–S composite, respectively. Manthiram et al. developed a flexible S-based cathode by loading S in N-doped DHCSs followed by graphene oxide wrapping [1]. The free-standing nanostructured sulfur cathode electrode without any binder enables a high discharge specific capacity of 1360 mA h g^−1^ at a current density of 0.2 C, an excellent rate capability of 600 mA h g^−1^ at a high current density of 2 C, and a long-cycling stability. 

Different from the hollow structure, the yolk–shelled structure can provide an improved electrochemical performance because of their unique buffering space, and short diffusion distance [185]. Wang et al. successfully designed the mesoporous N-doped yolk-shelled carbon (NYSC) nanospheres as novel S hosts [186]. From the TEM image, the yolk-shelled structure and a void gap between the exterior shell and the interior yolk can be clearly seen in Figure 13a. The shell thickness is approximately 140 nm, and the average diameter of the interior spherical yolk is approximately 410 nm. Figure 13b shows the typical charge and discharge curves of the NYSC@S electrode at a current density of 0.2 C in the 300th cycle, and the capacity is stable at 961 mA h g^−1^. The long-cycling performance and CE of the NYSC@S electrode at 0.2 C is shown in Figure 13c. The initial discharge specific capacity of NYSC@S electrode is 1329 mA h g^−1^. Even after 500 cycles, the capacity of NYSC@S is 909 mA h g^−1^ maintained. Wang et al. prepared the multi-shelled hollow carbon spheres (MHCS) and encapsulated a high percentage of S (86 wt%) loading through an in situ sulfur impregnation [187]. In the TEM image of the MHCS-S composites in Figure 13d, the diameter is approximately 150 nm, while the inset image in Figure 13d exhibits a shell thickness of 20 nm. Figure 13e shows that the MHCS-S composite possessed high discharge specific capacities of 1350 and 1003 mA h g^−1^ at the current densities of 0.1 and 1 C, respectively. Even after 200 cycles, the discharge specific capacities of the MHCS-S are 1250 and 846 mA h g^−1^ at the current densities of 0.1 and 1 C (Figure 13f), respectively. The high discharge specific capacity, good rate performance, and good capacity retention ensure the utilization of the MHCS structures in S hosting for LSBs. This strategy provides new ideas for the development of LSBs and even other MSHSs materials of electrode material optimization.

Carbon materials, especially those with simple configurations, provide insufficient confinement to immobilize polar polysulfides during the operation process. Compared with the hollow carbon sphere, the yolk–shelled and multi-shelled carbon structures are of special interest because of their improved confinement ability, large contact area with sulfur, and short transport length for Li^+^. Therefore, the introduction of suitable hosts which can facilitate the charge transport and confine the polysulfides plays a pivotal role in boosting the performance of LSBs. 

### 3.3. Supercapacitors

SCs are widely used as the electrochemical energy storage devices because of their long cycle life and high power density [188,189,190,191]. Based on energy storage mechanism, SCs can be divided into the following categories: EDLC caused by the charge accumulation in the electrode/electrolyte interface [192,193,194,195,196], pseudocapacitor based on the fast and reversible redox reactions at electrochemically active sites [197,198,199,200,201], and hybrid capacitors combining the electric double layer or pseudocapacitive active material with the battery active material [202,203,204,205,206,207]

Tremendous research efforts have been devoted to the design of nanostructured electrodes, especially hollow nanostructured electrodes with shortened diffusion lengths for ion transport and robust architectures for extended cycling capability. Several related studies are shown in Figure 14 and Figure 15. The NiCo_2_S_4_ hollow microspheres (Figure 1f) were synthesized through a hydrothermal method and were used as the SCs cathode material [74]. Figure 14a illustrates the progress of synthesizing the NiCo_2_S_4_ hollow nanospheres and rGO/Fe_2_O_3_. Then, NiCo_2_S_4_ was used as cathode material, and rGO/Fe_2_O_3_ was used as anode material. NiCo_2_S_4_ microspheres exhibit good rate capability from the galvanostatic charge and discharge (GCD) profiles in Figure 14b and excellent long-cycling performance with 91.5% retention of the initial capacitance after 1000 cycles (Figure 14c). The hollow structure enhances the structural stability, thereby causing excellent electrochemistry performance. Yan et al. reported the synthesis of the CoS_2_ solid spheres, yolk-shelled spheres, and SSHSs structures [109]. When evaluated as an electrode material for SCs, the CoS_2_ SSHSs delivered substantially improved capacitance and cycling performance over their solid and yolk-shelled spheres. 

Recently, our group reported the synthesis of core–shelled Ni-CeO_2_@PANI nanospheres (Figure 14d) with controlled amount of HCl [90]. Figure 2b shows that the Ni-CeO_2_@PANI nanospheres are composed of Ni-CeO_2_ core and PANI shell. As an electrode material, this material exhibits high specific capacitance of 866 F g^−1^ at a current density of 1 A g^−1^ (Figure 14e) and excellent cycling performance of 85.6% of the remaining content after 10,000 cycles (Figure 14f). Lou et al. demonstrated an asymmetric supercapacitor (ASC) device by using the NiCo_2_S_4_ DSHSs (Figs. 6o and p) as electrode material [121]. As shown in Figure 14g, in this system, the NiCo_2_S_4_ DSHSs was used as cathode material, graphene/C spheres (G/CSs) was used as anode material, and cellulose film was used as the separator in the KOH electrolyte (Figure 14g). Figure 14h shows that this cathode material of the ASC exhibits excellent cycling stability with only 20% loss of the initial specific capacitance over 10000 cycles. Moreover, this ASC displays an energy density of 22.9 Wh kg^−1^ even at a high power density of 10208 W kg^−1^ (Figure 14i), which is superior to many other ASCs that are previously reported.

Interestingly enough, researchers are constantly trying to come up with new strategies to help SCs better fit into real-life applications. The recent advances in the synthesis of complex hollow structures have provided opportunities to further optimize the performance of SCs. Here, we highlight three works with innovation and development potential. For example, Wang et al. synthesized the thin NiCo_2_O_4_ SSHSs, DSHSs, and TSHSs by controlling the penetrated amount of Ni^2+^ and Co^2+^ [130]. The TEM image (Figure 15a) demonstrates that the diameter of NiCo_2_O_4_ TSHSs is ~200–300 nm and has an average thickness of approximately 24 nm. The specific capacitance (Figure 15b) of the thin NiCo_2_O_4_ TSHSs reached 68 F g^−1^ at a high current density of 1 A g^−1^, which is maintained 41 F g^−1^ even at a high current density of 10 A g^−1^, demonstrating its excellent rate performance. Furthermore, Figure 15c shows the cycling performance of the ASCs with 77% capacitance retention after 1700 cycles at a 5 A g^−1^. Li et al. reported the fabrication of uniform the CoFe_2_O_4_ SSHSs, DSHSs, TSHSs, and QSHSs, which were evaluated as electrodes for SCs [139]. Figure 15d shows the TEM image of the CoFe_2_O_4_ TSHSs. Figure 15e shows the rate capability for these hollow electrodes at different current densities. The initial capacitances of the CoFe_2_O_4_ SSHSs, DSHSs, TSHSs, and QSHSs are 406.8, 552.8, 1450.0, and 1211.0 F g^−1^, respectively. Moreover, the CoFe_2_O_4_ TSHSs shows that promising cycle stability was approximately 98% retention after 500 cycles at a sweep scan rate of 50 mV s^−1^, indicating an excellent electrochemical performance (Figure 15f). Cao et al. reported a novel strategy for the controlled synthesis of the Co_3_O_4_ MSHSs [66]. The TEM image in Figure 15g demonstrates the MSHSs structures. MSHSs structures, including an exterior shell and two interior shells, are clearly detected with diameters of approximately 1.8, 1.0, and 0.5 μm, respectively. Moreover, the large void between the exterior shell and two interior shells is clearly exhibited. When evaluated for the SCs performance, the Co_3_O_4_ MSHSs exhibit high specific capacitances of 394.4 and 360 F g^−1^ at current densities of 2 and 10 A g^−1^ (Figure 15h), respectively. Figure 15i shows the long cycling stability of the Co_3_O_4_ MSHSs electrode was 92% retention of its original specific capacitance after 500 cycles even at the high current density of 2 A g^−1^.

Compared with the simple hollow sphere structures, complex hollow structures with more electroactive sites are expected to deliver higher electrochemical activity. The unique double-shelled structures may confine electrolyte between shells, providing a large driving force for electrochemical reactions. Furthermore, multi-shelled structures are believed to offer exceptional structural robustness for enhanced electrochemical stability. Hence, complex hollow structures are expected to be the next generation of the most promising SCs electrode materials.

## 4. Conclusions

Here, we have summarized the synthetic approaches and examples of energy storage related applications of micro/nanostructured spherical materials. The advances in the synthesis of these micro/nanostructured spherical materials have promoted their energy-related applications and highlighted their promising applications in LIBs, LSBs, and SCs. Compared with the simple spherical structures, the complexity of the material structures possess increased opportunities to adjust their performance, thereby contributing to improve the electrochemistry performance of the electrode materials. For the sulfur hosts for LSBs, and electrode materials for LIBs and SCs, these micro/nanostructured spherical materials show excellent electrochemical performance with high discharge specific capacity or capacitance and long life span cycling stability due to their high structural stability. 

In spite of these progresses, precise control and manipulation of these intricate hollow structures materials still need further investigating. From the synthetic viewpoint, the spherical structure materials with controllable dimension, morphology, complex structures, shell numbers, and desired compositions are hardly obtained through simple and convenience methods. The formation mechanism of some intricate hollow structures remains elusive. 

For future research in the synthesis of intricate hollow structures, we believe that it should concentrate on the following aspects: (1) further expanding and modifying the existing templating and template-free methods for complex hollow structures; (2) the combination of different methods will be a prevalent trend for synthesis of some special complex hollow structures, where multiple hollowing strategies can be involved; and (3) understanding of the synthesis mechanism of the MSHSs structures is conducive to its development and expansion of its application. 

Lastly, it should be emphasized that the syntheses and energy applications of complex hollow structures are still in their infancy. There is still a long way to go for the commercial-scale production and practical application of such intriguing materials. We are confident that other versatile and powerful synthetic methodologies for hollow structures will be developed soon.

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
