# Peer review of "Synthesis and Electrochemical Energy Storage Applications of Micro/Nanostructured Spherical Materials"

_nanomaterials, 2019, doi:10.3390/nano9091207_

Round 1
Reviewer 1 Report
The review-article from Zhou and co-workers deals with micro/nano-structured materials based on their synthetic preparation, as well as applicability for energy storage purposes.
Title:
Remove title and use micro/nanostructured instead of micro-…
Abstract:
The abstract should not only list the different topics covered in the article, but give the interested reader a short idea which concepts and contents are covered by the review article.
Introduction:
The introduction so far just itemizes all the concepts to be covered in the review article to introduce the necessary references. The same accounts also for the applicational part at the end of the introduction. For an introduction I would expect a detailed introduction in the field, maybe also with a short historical overview on the achievements until today. However, in the present form even for experts in the field it is hard to understand what will be covered in the review article, as no sufficient stage for the following part is set. Scheme 1 is misleading, as the authors seem to subdivide micro/nano structured materials into three groups, which than are further split into sub-groups. But in the final sub-groups the same material would appear more times, as in scheme 1 application is mixed with structure and synthesis.
Body:
I miss a clear structure for all the different sub-chapters. The authors give a very brief, slightly generalized introduction to the topic of the sub-chapter and continue than with a description of different literature examples. The large grouped figures are very hard to read and correlate with the description in the text, also because some of the representations are very small and of bad quality.
I would consider the authors to rework all the different sub-chapters in detail, taking care to i) give a detailed introductive description of the concept (not for only one paragraph) to summarize and highlight the peculiarities, advantages, disadvantages, potential etc. of the different synthetic methods, structures, etc. ii) Describe the most important achievements related to the topic of the sub-chapter and iii) Conclude each sub-chapter with a brief outlook / comment of the evolvement of the field in this regard.
Especially for the selection of the examples it is utmost necessary to explain why exactly this work is considered as one of the most interesting and prominent results in the field to avoid the impression of a rather randomized selection.
Conclusion:
I like the concept of stating the challenges encountered in the field. Nevertheless, the prior itemized repetitive list of covered concepts is superfluous and not suitable for the conclusion. A general conclusion from the perspective of the authors – being experts in the field – and giving their interpretation on the development, highlights and future directions of the field would be recommended.
Author Response
Professor Alisa Zhai,
Assistant Editor
Nanomaterials
Dear Prof. Zhai,
We would like to thank you and reviewers for your comments and suggestions. Our manuscript (nanomaterials-554986) has been revised according to the reviewers’ comments. The corresponding changes have been marked in red in the text. Our point-by-point responses to the reviewers’ comments are as follows.
Sincerely yours,
Guowei Zhou
Response to Reviewer 1 Comments
Point 1: Title: Remove title and use micro/nanostructured instead of micro-
Response 1: We thank the reviewer for the comments. According to the reviewer’s suggestion, in title the “micro-nanostructured” was revised to “micro/nanostructured”, and all corresponding contents in text were revised.
Point 2: Abstract: The abstract should not only list the different topics covered in the article, but give the interested reader a short idea which concepts and contents are covered by the review article.
Response 2: We thank the reviewer for the comments. According to the reviewer’s suggestions, the corresponding contents “the concepts and syntheses of micro/nanostructured spherical materials are described in detail” and “and also include the concepts of synthetic methodologies.” were newly added in Abstract.
The specific concepts were added in Body:
“Hollow spheres are simple spherical structures with narrow size distribution and superior morphological uniformity, and usually obtained through template method, such as hard- and soft-templating.” was added in Page 2 Line 73.
“Typically, double-shell micro/nanostructured spherical materials often possess double shells, hollow core, and a gap or no gap between the double shells, which are shown in Figure 6.” was added in Page 10 Line 321.
“Multi-shelled hollow spheres (MSHSs) include multi-shells and a hollow chamber. Despite the challenges brought by the high structural complexity of these multi-shelled hollow spheres, similar to single-shelled ones, they can also be synthesized based on well-controlled templating or self-templated methods. In this section, the synthetic methodologies for MSHSs, including well-established hard-, soft-, and self-templating methods, will be reviewed.” were added in Page 12 Line 400.
“Hard-templating method is facile and straightforward theoretically, which is firstly used to prepare MSHSs structures. In general, the hard-templating synthesis involves four major steps: i) template preparation; ii) surface modification of the hard template, iii) target material coating/deposition; and iv) template removal. The coating/deposition of the target material on a hard template is generally considered as the most challenging step. Sometimes, the surface modification can be omitted if the target material is compatible with the template. The most frequently employed hard templates include monodisperse polymer, silica, carbon, metal, and metal oxide colloids.” were added in Page 12 Line 406.
Point 3: Introduction:The introduction so far just itemizes all the concepts to be covered in the review article to introduce the necessary references. The same accounts also for the applicational part at the end of the introduction. For an introduction I would expect a detailed introduction in the field, maybe also with a short historical overview on the achievements until today. However, in the present form even for experts in the field it is hard to understand what will be covered in the review article, as no sufficient stage for the following part is set. Scheme 1 is misleading, as the authors seem to subdivide micro/nano structured materials into three groups, which than are further split into sub-groups. But in the final sub-groups the same material would appear more times, as in scheme 1 application is mixed with structure and synthesis.
Response 3: We thank the reviewer for the comments. According to the reviewer’s suggestions, the corresponding contents “In addition, LIBs can also be applied in various energy storage devices, including static storage media, portable electronic devices, and electric vehicles (EVs). In general, the configuration of LIBs composed of anode, cathode, separator, electrolyte, and coin cells [49–51]. During the process of charging and discharging, Li+ intercalate/deintercalate between anode and cathode through the electrolyte [52–54]. The performance of LIBs mainly depends on the anode materials, which are used to store and release Li+ during the process of charging and discharging [55–57]. LSBs have attracted increasing attention due to their high theoretical specific capacity (1675 mA h g−1), and energy density (2600 Wh kg−1) [58,59]. In addition, S has many valuable characteristics, such as low cost and safety, equivalent weight, and environment-friendliness. SCs is an important energy storage equipment due to their long cycle performances and high power density, and that is include pseudocapacitors and electrical double layer capacitors (EDLC) [60–65]. In contrast, pseudocapacitors have higher specific capacitances than EDLC, which attracted much research interest [66–68]. SCs are divided into the following categories based on energy storage mechanism: EDLC caused by the charge accumulation in the electrode/electrolyte interface and pseudocapacitor based on the fast and reversible redox reactions at electrochemically active sites [69,70].” were revised to “LIBs were first commercialized in 1991 by Sony Corporation [49-52]. Currently, LIBs provide a voltage of the order of 4 V, and energy density ranging from 100 to 150 Wh kg−1. LIBs have three main application domains: portable electronics, electric vehicle, and stationary energy storage [53-55]. So far, the first usage is the best developed and the largest in terms of the number of units generated [56,57]. Li–S batteries hold tremendous potential as energy storage devices due to their high theoretical specific capacity (1675 mA h g−1), and energy density (2600 Wh kg−1) [58]. Since 2009, Li–S batteries have received increasing attention and are considered as one of the most promising candidates for next-generation rechargeable batteries. From the recent improvements in the Li–S system, it seems that the practical application of Li–S batteries is not far away [59]. The earliest SCs patent was filed in 1957. However, not until the 1990s did SCs technology begin to draw some attention, in the field of hybrid electric vehicles [60-62]. It was found that the main function of SCs could be to boost the battery in a hybrid electric vehicle providing the necessary power for acceleration, with an additional function being to recuperate brake energy [63-66]. Further developments have led to the recognition that SCs can play important roles in complementing batteries or fuel cells in their energy storage functions by providing back-up power supplies to protect against power disruptions [67]. As a result, the US Department of Energy has designated SCs to be as important as batteries for future energy storage systems [68]. Recent years, major progress have been yielded in the theoretical and practical research and development of SCs, as evinced by a large number of research articles and technical reports [69,70]. ” in Page 1 Lines 42-60.
And Scheme 1 was redrawn and the new Scheme 1 was offered in page 2.
Point 4: Body: I miss a clear structure for all the different sub-chapters. The authors give a very brief, slightly generalized introduction to the topic of the sub-chapter and continue than with a description of different literature examples. The large grouped figures are very hard to read and correlate with the description in the text, also because some of the representations are very small and of bad quality.
I would consider the authors to rework all the different sub-chapters in detail, taking care to i) give a detailed introductive description of the concept (not for only one paragraph) to summarize and highlight the peculiarities, advantages, disadvantages, potential etc. of the different synthetic methods, structures, etc. ii) Describe the most important achievements related to the topic of the sub-chapter and iii) Conclude each sub-chapter with a brief outlook / comment of the evolvement of the field in this regard.
Especially for the selection of the examples it is utmost necessary to explain why exactly this work is considered as one of the most interesting and prominent results in the field to avoid the impression of a rather randomized selection.
Response 4: We thank the reviewer for the comments. According to the reviewer’s suggestions, for the grouped figures, we give the explanations in the text and the figure captions:
“The mechanism of formation process, transmission electron microscopy (TEM) and scanning electron microscopy (SEM) images of hollow spheres are illustrated in Figure 1.” was added in in Page 2 Line 75.
“Figure 1. Schematic demonstration and graphical illustration of hollow spheres:” was added in Page 3 Line 98.
“Figure 2. Schematic demonstration and graphical illustration of core-shelled spheres:” was added in Page 5 Line 174.
“Figure 3. Graphical illustration of core-shelled spheres:” was added in Page 6 Line 207.
“For a better understanding of these structures, we will review these materials based on different structure types, which are shown in Figure 4 and 5.” was added in Page 6 Line 222 and Page 7 line 223.
“Figure 4. Schematic demonstration and graphical illustration of the single shell with single yolk spherical structure:” was added in Page 8 Line 274.
“Figure 5. Schematic demonstration and graphical illustration of the yolk–shells and yolks–shell structure:” was added in Page 9 Line 311.
“Typically, double-shell micro/nanostructured spherical materials often possess double shells, hollow core , and a gap or no gap between the double shells, which are shown in Figure 6.” was added in Page 10 Line 321.
“Figure 6. Schematic demonstration and graphical illustration of double–shell structure:” was added in Page 11 Line 370.
“Figure 7. Schematic demonstration and graphical illustration of multi–shells structure synthesized by hard-templating method:” was added in Page 13 Line 441.
“Therefore, only a few successful examples are obtained, which are shown in Figure 8.” was added in Page 13 Line 450.
“Figure 8. Schematic demonstration and graphical illustration of multi–shells structure synthesized by soft-templating method:” in Page 14 Line 452.
“Figure 9. Schematic demonstration and graphical illustration of multi–shells structure synthesized by self-templating method:” was added in Page 16 Line 516.
“Figure 10. TEM images of hollow and double-shell spheres and their LIBs performances:” was added in Page 18 Line 592.
“Figure 11. TEM images of yolk-shell and multi-shell spheres and their LIBs performances:” was added in Page 19 Line 614.
“Figure 12. TEM images of hollow and double-shell spheres and their LSBs performances:” was added in Page 20 Line 665.
“Figure 13. TEM images of yolk-shell and multi-shell spheres and their LSBs performances:” was added in Page 21 Line 706.
“Tremendous research efforts have been devoted to the design of nanostructured electrodes, especially hollow nanostructured electrodes with shortened diffusion lengths for ion transport and robust architectures for extended cycling capability. Several related studies are shown in Figure 14 and 15.” were added in Page 22 Line 734.
“Figure 14. Schematic demonstration of hollow, core-shell, and double-shell spheres and their SCs performances:” was added in Page 22 Line 724.
“Figure 15. TEM images of triple-shell and multi-shell spheres and their SCs performances:” was added in Page 23 Line 762.
For 2.1. Hollow Spheres with Complex Architectures:
The corresponding content “Hollow spheres is a simple spherical structures, and usually obtained through template method, such as hard- and soft-templating. In contrast, hard-templating is the common method, and that can be selectively removed through etching or annealing. SiO2 and polystyrene (PS) are the most commonly used hard template owing to their facile preparation [71, 72].” was revised to “Hollow sphere is simple spherical structures with narrow size distribution and superior morphological uniformity, and usually obtained through template methods, such as hard- and soft-templating.” in Page 2 Line 73.
The corresponding contents “Hard-templating is the common method, and the hard template can be selectively removed through etching or annealing. Typically, the desired materials or their precursors are deposited on hard templates with functionalized surface, followed by the selective removal of the templates through etching or pyrolysis. A myriad of inorganic/organic colloidal spheres could be applied as ideal hard templates (SiO2, polystyrene (PS), and so on) owing to their facile preparation [71, 72].” were added in Page 3 Line 78.
The corresponding contents “For soft-templating methods, the involved template (emulsion droplets, micelles, vesicles, microemulsion, and gas bubbles) are generally in the form of fluid/gas with high deformability. Thus, the complicated template elimination process is generally not necessary [75,76].” were added in Page 3 Line 105.
The corresponding contents “For the synthetic methods of the hollow spheres, the hard template can be selectively etched, whereas the complicated template elimination process of soft template is generally not necessary. With this method, product uniformity is sometimes compromised. However, the possibility of producing more complicated hierarchical structures is largely increased by refilling a hollow interior with functional species or the in-situ encapsulation of guest molecules during shell formation. Therefore, the soft template method is more suitable for the preparation of hollow spheres.” were newly added in Page 4 Line 133.
For 2.2. Core-shelled Spheres with Complex Architectures:
The corresponding contents “For the one solid inner core coated with one shell material,” and “For the one solid inner core coated with two or more shells materials,” were added in Page 4 Line 147 and Page 5 Line 182, respectively.
The corresponding contents “The main advantages of these core–shell structures include the following ability to: 1) protect the core from the effect of environmental changes outside; 2) intensify or introduce new chemical or physical capabilities; 3) limit volume expansion and maintain structural integrity; 4) protect the core from aggregating into large particles; and 5) percolate ions or molecules onto the core selectively.” was added in Page 6 Line 211.
For 2.3. Yolk-shelled Spheres with Complex Architectures:
The corresponding contents “Yolk–shelled structure materials were first synthesized through silica template by Hyeon et al. Initial researches of yolk–shelled structures concentrated on spherical structures. For a better understanding of these structures, we will review these materials based on different structure types, as shown in Figure 4 and 5.” was added in Page 6 Line 221.
The corresponding content “Apart from a single shell with single yolk, yolk–shells and yolks–shell structure were reported, which are shown in Figure 5.” was added in Page 8 Line 281.
The corresponding contents “Typical spherical yolk–shelled structures are tuned with various numbers of shells and yolks. The suitable void space between yolk and shell can accommodate the volume expansion of yolk and avoid aggregation of electroactive cores during charging/discharging process. With the development of different synthetic methods, yolk–shelled structures can be prepared into manifold types.” were added in Page 10 Line 316.
For 2.4. Double-shelled Spheres with Complex Architectures:
The corresponding content “Typically, double-shell micro/nanostructured spherical materials often possess double shells, hollow core, and a gap or no gap between the double shells, which are shown in Figure 6.” was added in Page 10 Line 321.
The corresponding contents “For the double-shell micro/nanostructured spherical materials, more synthetic methods such as LBL growth, sol-gel method, hard template method, soft template method, and ion exchange approaches have been reported. LBL growth and hard template method are simple, effective, and straightforward in concept, whereas the soft template method generally be free from the complicated template elimination process. A suitable void space in inner or between the double shells can accommodate the volume expansion, increase the specific surface area of the material, and increase the contact area between the material and the electrolyte. In order to further increase the specific surface area of the material, the mesoporous features could be introduced to double-shell structures.” were added in Page 12 Line 391.
For 2.5. Multi-shelled Spheres with Complex Architectures:
The corresponding contents “Multi-shelled hollow spheres (MSHSs) include multi-shells and a hollow chamber. Despite the challenges brought by the high structural complexity of these multi-shelled hollow spheres, similar to single-shelled ones, they can also be synthesized based on well-controlled templating or self-templated methods. In this section, the synthetic methodologies for MSHSs, including well-established hard-, soft-, and self-templating methods, will be reviewed.” were added in Page 12 Line 400.
The corresponding contents “Hard-templating method is facile and straightforward theoretically, which is firstly used to prepare MSHSs structures. In general, the hard-templating synthesis involves four major steps: i) template preparation; ii) surface modification of the hard template, iii) target material coating/deposition; and iv) template removal. The coating/deposition of the target material on a hard template is generally considered as the most challenging step. Sometimes, the surface modification can be omitted if the target material is compatible with the template. The most frequently employed hard templates include monodisperse polymer, silica, carbon, metal, and metal oxide colloids.” were added in Page 12 Line 406.
The corresponding contents “Hard templates are monodispersity, easy size and shape controllable, ready availability in large amounts, and easy synthesis using well-known recipes. In spite of these advances, hard-templating methods still face quite a few challenges, such as the difficulty to achieve uniform coating due to compatibility issues between the templates and desired shell materials, and the tedious template removal processes. However, the MSHSs structure is easily influenced by pH, temperature, solvent, and ionic strength. Unlike conventional hard-/soft-templating approaches, self-templating synthesis enables good control over the particle uniformity without an auxiliary template removal process, which simplifies synthetic procedures, reduces production costs, and provides ease for scale up. Hence, modified templating strategies with an extra conversion step could transfer compatible precursor shells into target materials. Through these synthetic protocols, core and shell substances are elaborately selected to avoid potential compatibility problems.” were added in Page 17 Line 538.
For 3.1. Lithium-Ion Batteries:
The corresponding contents “Although it possesses above mentioned advances, simple hollow nanostructures can only provide limited possibilities to modulate the properties of electrode materials. Therefore, further manipulation of hollow structures in terms of geometric morphology, chemical composition, and shell architecture for complex architectures is required to achieve improved electrochemical performance demanded by the current emerging technologies.” were added in Page 18 Line 599.
The corresponding contents “Improved lithium storage performance has been realized by simultaneously manipulating the morphology of hollow particles and their compositions. Rational incorporation of multi-compositions in the shells of hollow nanostructures might combine the advantages of different materials, and the synergistic effect arising from their interaction promises improvement for lithium storage performance. The multi-shelled hollow structures can improve the volumetric energy density of electrodes by increasing the weight fraction of the active species, and also extend the cycle life due to the enhanced structural stability.” were added in Page 20 Line 648.
For 3.2 Lithium–Sulfur Batteries:
The corresponding contents “Carbon materials, especially for those with simple configurations, provide insufficient confinement to immobilize polar polysulfides during the operation process. Compared with the hollow carbon sphere, the yolk–shelled and multi-shelled carbon structures are of special interest because of their improved confinement ability, large contact area with sulfur, and short transport length for Li+. Therefore, the introduction of suitable hosts which can facilitate the charge transport and confine the polysulfides plays a pivotal role in boosting the performance of LSBs.” were added in Page 21 Line 710.
For 3.3. Supercapacitors:
The corresponding contents “Tremendous research efforts have been devoted to the design of nanostructured electrodes, especially hollow nanostructured electrodes with shortened diffusion lengths for ion transport and robust architectures for extended cycling capability. Several related studies are shown in Figure 14 and 15.” were added in Page 22 Line 734.
The corresponding content “The recent advances in the synthesis of complex hollow structures have provided opportunities to further optimize the performance of SCs.” was added in Page 23 Line 768.
The corresponding contents “Compared with the simple hollow sphere structures, complex hollow structures with more electroactive sites are expected to deliver higher electrochemical activity. The unique double-shelled structures may confine electrolyte between shells, providing a large driving force for electrochemical reactions. Furthermore, multi-shelled structures are believed to offer exceptional structural robustness for enhanced electrochemical stability. Hence, complex hollow structures are expected to be the next generation of the most promising SCs electrode materials.” were added in Page 24 Line 791.
The examples mentioned in this review may not be one of the most interesting and prominent results in the field, but they are consistent with the structural classification or energy-related applications of these materials and are representative.
Point 5: Conclusion: I like the concept of stating the challenges encountered in the field. Nevertheless, the prior itemized repetitive list of covered concepts is superfluous and not suitable for the conclusion. A general conclusion from the perspective of the authors–being experts in the field–and giving their interpretation on the development, highlights and future directions of the field would be recommended.
Response 5: We thank the reviewer for the comments. According to the reviewer’s suggestions, the corresponding contents in conclusion were revised to “In spite of these progresses, precise control and manipulation of these intricate hollow structures materials still need further investigating. From the synthetic viewpoint, the spherical structure materials with controllable dimension, morphology, complex structures, shell numbers, and desired compositions are hardly obtained through simple and convenience methods. And the formation mechanism of some intricate hollow structures remains elusive.
For future research in the synthesis of intricate hollow structures, we believe that it should concentrate on the following aspects: 1) further expanding and modifying the existing templating and template-free methods for complex hollow structures; 2) the combination of different methods will be a prevalent trend for synthesis of some special complex hollow structures, where multiple hollowing strategies can be involved; and 3) understanding of the synthesis mechanism of the MSHSs structures is conducive to its development and expansion of its application.
Lastly, it should be emphasized that the syntheses and energy applications of complex hollow structures are still in its infancy. There is still a long way to go for the commercial-scale production and practical application of such intriguing materials. We are confident that other versatile and powerful synthetic methodologies for hollow structures will be developed soon. Moreover, intricate hollow structures will break some of the current bottlenecks in the fields of energy storage and conversion.”.
Reviewer 2 Report
The review focuses on the micro-structured spherical materials and their application in electrochemical devices. The authors categorized the different types of spherical structures on the basis of their complexity: hierarchical hollows, core-shelled, yolk-shelled, double and multi shelled spheres. They provide the design and synthesis approaches to prepare microspheres with improved performances as electrodes in Lithium-Ion batteries, Lithium–Sulfur batteries, and Supercapacitors.
The subject is appropriate for “Nanomaterials”. The review is really interesting: while summarizing the different synthetic routes to prepare micro-nanostructures spherical materials with peculiar and complex morphologies, it put into evidence their chances as electrode materials for improved electrochemical performances.
The paper is well structured and the content is well developed. I think the paper can by published in “Nanomaterials” after minor revision. The suggested revision mainly concerns the English language: it does not always meet the standard for publications, and some sentences (here after reported) do not sound:
Page 2
Line 59: “Their applications the sulfur hosts for LSBs, electrode materials”
Lines 68-69: “In contrast, hard-templating is the common method, and that can be selectively removed through etching or annealing.”
Page 4
Lines 142-144: “From that, the MoS2 nanosheets grown evenly on the TiO2 surfaces, and the average diameter of TiO2@MoS2 is approximately 580–143 nm, and the TEM image displays the thickness of the MoS2 shells is around 130–170 nm (Figure 144 2f).”
Page 8
Lines 229-230: “The magnified TEM image in Figure 4f can be clearly found tne inner RF-protected magnetic Fe3O4 core.”
Page 10
Lines 304-305: “Figure 6c and d show the FESEM and TEM images of the SnO2@C DSHSs, that can be clearly seen the DSHSs structure.”
Page 11
Line 326: “carbonization again”
Lines 343-344: “In Figure 6m, that can be clearly seen the diameter of the inner cavity size is approximately 250 nm, and the thickness of the inner and outer shells is approximately 90 nm.”
Lines 360-361: “Multi-shelled hollow spheres (MSHSs) are included multi-shells and a hollow chamber. And their can be synthesized by hard-, soft-, or self-templating methods.”
Line 366: “grown”
Page 14
Lines 446-447: “This strategy without the procedure of template removal, which simplifies the synthetic processes, and decreases the production costs [145].”
Lines 453-454: “Subsequently, the Ce3+ ions adsorption with the carbon microspheres in an alkaline environment through electrostatic attractions.”
Page 16
Line 515: “that can be confirmed that...”
Page 18
Lines 564-565: “As shown in Figure 11j, the first cycle discharge and charge curves of the samples show that the Co3O4 DSHSs with the highest discharge specific capacity.”
Page 19
Lines 592-594: “suggesting that even the core part of the composite material can be readily contacted the electrolyte. Due to the unique structures and compositional advantages, a better capacity retention of 65.5% over 500 cycles at 0.5 C (Figure 12c).”
Page 20
Lines 624-625: “Even after 200 cycles, the discharge specific capacities of the MHCS-S were 1250 and 846 mA h g−1 can be retained at the current densities of 0.1 and 1 C (Figure 13f), respectively.”
Page 21
Lines 674-676: “The TEM image (Figure 15a) demonstrates that the NiCo2O4 TSHSs with a diameter of ~200–300 nm and an average thickness of approximately 24 nm.”
Typos
Page 2
line 56: change “include” into “included”
line 67: change “is a” into “are”
Page 4
Line 108: change mm into micrometers
Page 8:
Line 243: change “a large spaces” onto “a large space”
Line 263: change “to synthesis” into “to synthesize”
Page 16
Line 480: change “to synthesis” into “to synthesize”
Line 334: change “with a cavity diameter is approximately” into “with a cavity diameter approximately”
Page 17
Line 546: change “could effected” into “could affect”
Line 558: change “recover” into “recovers”
Page 19
Line 600: change “ffree” into “free”
Page 22
Line 717: change “are still have” into “still have”
Author Response
Professor Alisa Zhai,
Assistant Editor
Nanomaterials
Dear Prof. Zhai,
We would like to thank you and reviewers for your comments and suggestions. Our manuscript (nanomaterials-554986) has been revised according to the reviewers’ comments. The corresponding changes have been marked in red in the text. Our point-by-point responses to the reviewers’ comments are as follows.
Sincerely yours,
Guowei Zhou
Response to Reviewer 2 Comments
Point 1: Page 2 Line 59: “Their applications the sulfur hosts for LSBs, electrode materials”
Response 1: We thank the reviewer for the comments. According to the reviewer’s suggestion, the corresponding content was revised to “Their applications as the sulfur hosts for LSBs, electrode materials for LIBs and SCs conversion reactions are then discussed thoroughly.” in Page 2 Line 64.
Point 2: Lines 68-69: “In contrast, hard-templating is the common method, and that can be selectively removed through etching or annealing.”
Response 2: We thank the reviewer for the comments. According to the reviewer’s suggestion, the corresponding content was revised to “Hard-templating is the common method, and the hard template can be selectively removed through etching or annealing.” was added in Page 2 Line 78.
Point 3: Page 4 Lines 142-144: “From that, the MoS2 nanosheets grown evenly on the TiO2 surfaces, and the average diameter of TiO2@MoS2 is approximately 580–143 nm, and the TEM image displays the thickness of the MoS2 shells is around 130–170 nm (Figure 2f).”
Response 3: We thank the reviewer for the comments. According to the reviewer’s suggestion, the corresponding content was revised to “The MoS2 nanosheets grown evenly on the TiO2 surface and the average diameter of TiO2@MoS2 is approximately 580–620 nm. The TEM image displays the thickness of the MoS2 shells in the range of 130–170 nm (Figure 2f).” in Page 4 Lines 160-162
Point 4: Page 8 Lines 229-230: “The magnified TEM image in Figure 4f can be clearly found tne inner RF-protected magnetic Fe3O4 core.”
Response 4: We thank the reviewer for the comments. According to the reviewer’s suggestion, the corresponding content was revised to “The magnified TEM image in Figure 4f can be clearly found the inner RF-protected magnetic Fe3O4 core.” in Page 7 Line 248.
Point 5: Page 10 Lines 304-305: “Figure 6c and d show the FESEM and TEM images of the SnO2@C DSHSs, that can be clearly seen the DSHSs structure.”
Response 5: We thank the reviewer for the comments. According to the reviewer’s suggestion, the corresponding content “Figure 6c and d show the FESEM and TEM images of the SnO2@C DSHSs, that can be clearly seen the DSHSs structure.” was revised to “The DSHSs structure of the SnO2@C can be clearly obtained in Figure 6c and d.” in Page 10 Line 337.
Point 6: Page 11 Line 326: “carbonization again”
Response 6: We thank the reviewer for the comments. According to the reviewer’s suggestion, the corresponding content “Calcined again, the p–n heterostructured TiO2/NiO DSHS was obtained.” was revised to “The p–n heterostructured TiO2/NiO DSHS was obtained after the annealing process.” in Page 12 Line 350.
Point 7: Lines 343-344: “In Figure 6m, that can be clearly seen the diameter of the inner cavity size is approximately 250 nm, and the thickness of the inner and outer shells is approximately 90 nm.”
Response 7: We thank the reviewer for the comments. According to the reviewer’s suggestion, the corresponding content was revised to “In Figure 6m, it can be clearly seen that the diameter of the inner cavity size is approximately 250 nm, and the thickness of the inner and outer shells is approximately 90 nm.” in Page 11 Line 367.
Point 8: Lines 360-361: “Multi-shelled hollow spheres (MSHSs) are included multi-shells and a hollow chamber. And their can be synthesized by hard-, soft-, or self-templating methods.”
Response 8: We thank the reviewer for the comments. According to the reviewer’s suggestion, the corresponding contents were revised to “Multi-shelled hollow spheres (MSHSs) include multi-shells and a hollow chamber. Despite the challenges brought by the high structural complexity of these multi-shelled hollow spheres, similar to single-shelled ones, they can also be synthesized based on well-controlled templating or self-templated methods. In this section, the synthetic methodologies for MSHSs, including well-established hard-, soft-, and self-templating methods, will be reviewed.” in Page 12 Lines 400-404.
Point 9: Line 366: “grown”
Response 9: We thank the reviewer for the comments. According to the reviewer’s suggestion, the corresponding content “grown alternately” was revised to “are alternately grown” in Page 12 Line 415.
Point 10: Page 14 Lines 446-447: “This strategy without the procedure of template removal, which simplifies the synthetic processes, and decreases the production costs [145].”
Response 10: We thank the reviewer for the comments. According to the reviewer’s suggestion, the corresponding content was revised to “This strategy does not need to remove the template, which simplifies the synthetic processes and decreases the production costs [145]” in Page 15 Line 497.
Point 11: Lines 453-454: “Subsequently, the Ce3+ ions adsorption with the carbon microspheres in an alkaline environment through electrostatic attractions.”
Response 11: We thank the reviewer for the comments. According to the reviewer’s suggestion, the corresponding content was revised to “Subsequently, the Ce3+ ions are adsorbed onto the carbon microspheres in an alkaline environment through electrostatic attractions.” in Page 15 Line 504.
Point 12: Page 16 Line 515: “that can be confirmed that...”
Response 12: We thank the reviewer for the comments. According to the reviewer’s suggestion, the corresponding content was revised to “It can be confirmed that...” in Page 17 Line 578.
Point 13: Page 18 Lines 564-565: “As shown in Figure 11j, the first cycle discharge and charge curves of the samples show that the Co3O4 DSHSs with the highest discharge specific capacity.”
Response 13: We thank the reviewer for the comments. According to the reviewer’s suggestion, the corresponding content was revised to “As shown in Figure 11j, the first cycle discharge and charge curves of the samples show that the Co3O4 DSHSs have the highest discharge specific capacity.” in Page 19 Line 642.
Point 14: Page 19 Lines 592-594: “suggesting that even the core part of the composite material can be readily contacted the electrolyte. Due to the unique structures and compositional advantages, a better capacity retention of 65.5% over 500 cycles at 0.5 C (Figure 12c).”
Response14: We thank the reviewer for the comments. According to the reviewer’s suggestion, the corresponding content was revised to “suggesting that even the core part of the composite material can readily contact with the electrolyte. Due to the unique structures and compositional advantages, a better capacity retention is achieved to 65.5% over 500 cycles at 0.5 C (Figure 12c).” in Page 21 Line 674.
Point 15: Page 20 Lines 624-625: “Even after 200 cycles, the discharge specific capacities of the MHCS-S were 1250 and 846 mA h g−1 can be retained at the current densities of 0.1 and 1 C (Figure 13f), respectively.”
Response 15: We thank the reviewer for the comments. According to the reviewer’s suggestion, the corresponding content was revised to “Even after 200 cycles, the discharge specific capacities of the MHCS-S are 1250 and 846 mA h g−1 at the current densities of 0.1 and 1 C (Figure 13f), respectively.” in Page 21 Lines 701-702.
Point 16: Page 21 Lines 674-676: “The TEM image (Figure 15a) demonstrates that the NiCo2O4 TSHSs with a diameter of ~200–300 nm and an average thickness of approximately 24 nm.”
Response 16: We thank the reviewer for the comments. According to the reviewer’s suggestion, the corresponding content was revised to “The TEM image (Figure 15a) demonstrates that the NiCo2O4 TSHSs have a diameter of ~200–300 nm and an average thickness of approximately 24 nm.” in Page 23 Lines 771-773.
Point 17: Page 2 line 56: change “include” into “included”
Response 17: We thank the reviewer for the comments. According to the reviewer’s suggestion, the corresponding content “include” was revised to “included” in Page 2 Line 62.
Point 18: line 67: change “is a” into “are”
Response 18: We thank the reviewer for the comments. According to the reviewer’s suggestion, the corresponding content “Hollow spheres is a” was revised to “Hollow spheres are” in Page 2 Line 73.
Point 19: Page 4 Line 108: change “mm” into “micrometers”
Response 19: We thank the reviewer for the comments. According to the reviewer’s suggestion, in order to consistent with the full paper, the corresponding content “mm” was revised to “μm” in Page 4 Line 121.
Point 20: Page 8: Line 243: change “a large spaces” onto “a large space”
Response 20: We thank the reviewer for the comments. According to the reviewer’s suggestion, the corresponding content “a large spaces” was revised to “a large space” in Page 8 Line 262.
Point 21: Line 263: change “to synthesis” into “to synthesize”
Response 21: We thank the reviewer for the comments. According to the reviewer’s suggestion, the corresponding content “to synthesis” was revised to “to synthesize” in Page 9 Line 290.
Point 22: Page 16 Line 480: change “to synthesis” into “to synthesize”
Response 22: We thank the reviewer for the comments. According to the reviewer’s suggestion, the corresponding content was revised to “to synthesize” in Page 16 Line 532.
Point 23: Line 334: change “with a cavity diameter is approximately” into “with a cavity diameter approximately”
Response 23: We thank the reviewer for the comments. According to the reviewer’s suggestion, the corresponding content was revised to “with a cavity diameter of approximately” in Page 12 Line 358.
Point 24: Page 17 Line 546: change “could effected” into “could affect”
Response 24: We thank the reviewer for the comments. According to the reviewer’s suggestion, the corresponding content was revised to “could affect” in Page 19 Line 624.
Point 25: Line 558: change “recover” into “recovers”
Response 25: We thank the reviewer for the comments. According to the reviewer’s suggestion, the corresponding content was revised to “recovers” in Page 20 Line 636.
Point 26: Page 19 Line 600: change “ffree” into “free”
Response 26: We thank the reviewer for the comments. According to the reviewer’s suggestion, the corresponding content was revised to “free” in Page 21 Line 682.
Point 27: Page 22 Line 717: change “are still have” into “still have”
Response 27: We thank the reviewer for the comments. According to the reviewer’s suggestions, the corresponding content “In spite of these progresses, the synthesis methods and applications of these intricated hollow structures materials are still have some challenges needed to be overcome” was revised to “In spite of these progresses, precise control and manipulation of these intricate hollow structures materials still need further investigating.” in Page 25 Lines 808-809.
Round 2
Reviewer 1 Report
The authors worked on my comments, added a few helpful paragraphs, and also improved the scheme 1. Now the scheme is quite clear.
Nevertheless, some substantial comments were so far not addressed.
I would ask the authors to make clear in the article, why the single articles they refer to where selected.
Author Response
Response to Reviewer 1 Comments
Point 1: The authors worked on my comments, added a few helpful paragraphs, and also improved the scheme 1. Now the scheme is quite clear.
Nevertheless, some substantial comments were so far not addressed.
I would ask the authors to make clear in the article, why the single articles they refer to where selected.
Response 1: We thank the reviewer for the comments. According to the reviewer’s comments, the corresponding contents were added as follows:
“The examples mentioned in this review may not be one of the most interesting and prominent results in the field, but they are consistent with the structural classification or energy-related applications of these materials and are representative.” in page 2 lines 83-85.
“SiO2 spheres with controllable size can be easily prepared by the Stöber process on a large scale. Therefore, we select two typical examples of using SiO2 spheres as the hard templates.” in page 3 lines 105-107.
“Below, we introduce two interesting works with coconut-like and flower-like yolk–shelled spheres, respectively, using surfactant aggregation as templates.” in page 8 lines 283-284.
“As a specific example, Li et al. first reported the preparation of the anatase–rutile TiO2 double-shelled hollow spheres (DSHSs) through a facile sol-gel method using SiO2 nanospheres as the hard template [112].” in page 10 lines 349-351.
“First, some successful examples are reported by the author to prove that we have a full understanding of this synthesis method [133,134].” in page 14 lines 476-477.
“This is the first report using cyclodextrin as template for accurate synthesis of shell-controllable CoFe2O4 QSHSs.” in page 15 lines 504-506.
“Comparing with several vanadium based hollow materials, V2O3@C hollow spheres exhibit high reversible capacities as well as superior rate performance when they were used as anode materials for LIBs.” in page 18 lines 600-602.
“Its capacity is stable and its cycle stability is obviously improved. This is an effective strategy for promoting the electrochemical performance of LIBs.” in page 20 lines 666-667.
“This strategy provides new ideas for the development of LSBs and even other MSHSs materials of electrode material optimization.” in page 22 lines 735-736.
“Interestingly enough, researchers are constantly trying to come up with new strategies to help SCs better fit into real-life applications. The recent advances in the synthesis of complex hollow structures have provided opportunities to further optimize the performance of SCs. Here, we highlight three works with innovation and development potential.” in page 23 lines 793-796.
Reviewer 2 Report
The suggested minor corrections have been taken into account by the authors. I think the current version of the paper can be accepted for publication in “Nanomaterials” (Recommendation: accept).
Author Response
Point 1: The suggested minor corrections have been taken into account by the authors. I think the current version of the paper can be accepted for publication in “Nanomaterials” (Recommendation: accept).
Response 1: We thank the reviewer for the positive comment.
Round 3
Reviewer 1 Report
Some helpful comments were added
The authors are encouraged to strongly reconsider the statement
"The examples mentioned in this review may not be one of the most interesting and prominent results in the field [...]"
Author Response
Response to Reviewer 1 Comments
Point 1: The authors are encouraged to strongly reconsider the statement "The examples mentioned in this review may not be one of the most interesting and prominent results in the field [...]"
Response 1: We thank the reviewer for this suggestion. After carefully consideration, we realized that the statement above-mentioned is not suitable. We have revised this sentence as “The examples we enumerated in this review are the typical representatives in terms of the nano/micro-architectures related to the energy applications.” in the revised manuscript in Page 2 Lines 83-84.
